# Reviewing the Online Tourism Value Chain

**Carmen Berne-Manero \*, Maria Gómez-Campillo, Mercedes Marzo-Navarro and Marta Pedraja-Iglesias**

Marketing Department, University of Zaragoza, Facultad de Economía y Empresa, 2, 50005 Zaragoza, Spain; mgc@unizar.es (M.G.-C.); mmarzo@unizar.es (M.M.-N.); mpedraja@unizar.es (M.P.-I.)

\* Correspondence: cberne@unizar.es; Tel.: +34-976-761-000

**Abstract:** The booking purchase process in B2C tourism online from the perspective of the quality-satisfaction-loyalty value chain has scarcely been investigated. The measurement models of the variables are not unified and essential variables, as transaction costs, need more research in order to achieve a comprehensive model of the digital tourist purchase process. This research is aimed at solving this gap through the proposal of a theoretical structural model, which is tested for the Spanish context. The results show that the measurement of website-perceived quality must include utilitarian and hedonic aspects, which can provide a competitive advantage to acquire and retain customers. Perceived quality and transaction costs determine customer's satisfaction and, ultimately, repurchase intentions or brand loyalty. Prices are found as mediator variables fostering the effect of quality on satisfaction, and non-monetary costs act as a cause of satisfaction. The online B2C tourism business must implement efficient internal and external processes to justify perceived costs.

**Keywords:** B2C tourism online; online booking purchases; re-purchase intentions; satisfaction; transaction costs; value chain; website quality

## 1. Introduction

The latest research literature on tourism studies the evolution of virtual channels as a result of the intensive application of Information Technology and Communication (ICT). Since the establishment of Internet use initiated in the 90s (Oskan and Zandberg 2016), major changes have been observed in the consumers' purchase process. The relevance of paying attention to this evolution and consequences is that the customer purchase is implicit in the company's service-profit chain (Heskett et al. 2008).

The results of B2C tourism online companies depend on the successful accomplishment of the customers' purchasing process and on the right decisions in the distribution channels regarding the acquisition and retention of customers (end consumers in consumer markets). The emergence of online tourism channels has decreased the gap between providers and consumers (Ponte et al. 2015). The growing co-production or participation of the consumer in the production and distribution processes is considered as an initial stage of a continuum towards co-creation, which aims at the creation of innovations based on the collaboration of the client (Shaw et al. 2011). Consumers are essential as co-producers in the online tourism purchase processes, and companies need to ensure a suitable interaction with them and to adapt the characteristics of the service as much as possible. This task covers the whole process from the first electronic contact (website quality), to the purchase decision (website quality and satisfaction), and the effects on customer retention to the brand (loyalty).

This customer-centred strategic perspective follows the basic quality-satisfaction-loyalty chain, a sequence of the purchasing process that explains the results of an online tourism agent from the customer point of view. In the first phase of search and assessment, consumers gain a perception of the quality of the company's products (Kim and Lee 2005); when a

consumer chooses a certain e-provider, the process leads to a second phase of co-production that, when completed, results in a certain satisfaction level (e.g., Kim et al. 2011; Hung et al. 2014) or dissatisfaction (Amin and Nasharuddin 2013), and ultimately in some degree of loyalty in terms of intention to repurchase or in actual repetition behaviour (e.g., Ali et al. 2012; Ryu et al. 2012; Betancourt et al. 2017). There is some relevant research integrating the search-purchase-consumption process regarding the traditional service sector channels (e.g., Grewal et al. 1998; Olsen 2002; Darsono and Junaedi 2006; Helgesen et al. 2010; Ho et al. 2017). In particular, authors like Anderson et al. (1994); González et al. (2007); Yüksel et al. (2010); Chen and Chen (2010); Chen and Xiao (2013); Deng et al. (2013), and Gallarza et al. (2013), have provided interesting contributions to this knowledge area regarding the traditional tourism service distribution. However, there is not enough research that addresses the cause-effect relationships value chain in the case of the online purchasing process (Grissemann and Stokburger-Sauer 2012). The models are usually specified in the case of a particular level of the online distribution channel; they present some changes in the cause-effect structure, and they do not consider all the basic variables. The role of perceived transaction costs is particularly forgotten. Moreover, the measurement models of the variables are not unified.

Perceived quality is the most extensively researched by itself (see Law et al. 2010) but not so much as an antecedent of satisfaction (Hao et al. 2015), and rarely of loyalty (Park et al. 2007). Recently, Ali (2016) recognized the importance of investigating the complete sequence of website quality-satisfaction-loyalty and provided a model that confirms this structure of relationships for the hotel context.

The most striking lack is the shortage of research on the effect of the transaction costs—other than prices—incurred by the customer in the purchase process on his/her loyalty intentions. Due to the higher level of participation of customers required in the online purchase process, the transaction costs incurred by customers other than the price paid for the final product could have a significant role in determining the intentions to repurchase or brand loyalty.

These antecedents warn of the importance to deepen the research in order to provide a better understanding of the e-booking purchase process and a comprehensive model focused on the electronic B2C tourism. Thus, this research endeavours to cover the limitations observed through analysing the dynamics of the online tourism channel from the point of view of digital tourism customers and in accordance with the quality-satisfaction-loyalty scheme of the website. From the literature review of the relationships between the three milestones involved and from exploring the role of transaction costs within the value chain, the ultimate goal of this research is to provide an integral model of the relationships between the basic variables involved in the process of purchasing online tourist reservations. Thus, there are two research questions that summarize our research interests:

- Do the basic relationships of the quality-satisfaction-loyalty value chain validated in other contexts, work in the same way in online tourism?
- What is the role of the perceived transaction costs (relative to the customer's participation in the co-production of the online channel tourism service) in the value chain?

To tackle these questions, the first section of this work reviews the research literature to make clear the prior contributions and to justify the hypotheses and a theoretical model. The second section describes the empirical setting and the data collection procedure. The third section presents the results. From these results, useful conclusions and management implications are drawn as well as future research lines.

## 2. Literature Antecedents and Theoretical Model

The most common case in online purchase of tourism services is that the consumer purchases the right to use a tourism service in advance. The online tourism channel is where the purchase process takes place, separated to a great extent in time and space from the final product experience.

This separation brings the need to evaluate the booking purchase process per se. However, the crucial role of customer evaluation of the three steps in the purchase process—perceived quality, satisfaction and repurchase—is not encompassed with the attention paid in the context of online tourism purchase research.

*2.1. The Online Tourism Quality-Satisfaction-Loyalty Value Chain*

From the point of view of the customers, the performance indicators of a company in the market follow the quality-satisfaction-loyalty basic framework. In the current context of study, this framework regards the digital tourist purchase process that explains the results sequence of an online tourism agent (online tourism brand).

Before deciding on an online booking purchase, consumers try to match their quality expectations with the perceived quality of the product-service offered by online tourism agents, including suppliers, searchers, meta-searchers and online travel agencies as retailers. Balanced perceptions could lead them to make the decision to purchase with the consequent effects on the results of a market-oriented company (Grissemann and Stokburger-Sauer 2012).

A website is the main communication channel between service suppliers and consumers (Ali 2016). Ruiz-Mafe et al. (2018) affirm that when booking tourism products in an online community, consumers are generally unable to make valued judgements prior to purchase because of the lack of information regarding product quality. Nevertheless, when it deals with customers which purchase through the website of a touristic provider or an intermediary, the website quality is evaluated by the user through different attributes. Perceived quality is a parameter used to evaluate the performance of tourism organizations, destinations, hotels and travel agencies (Hao et al. 2015). Website quality captures the users' evaluation of whether a website's features meet their needs and reflects the overall excellence of the website (Chang and Chen 2008). Wang et al. (2015) compare the website as an online store where users need to rely on its attributes to reach a purchase decision. Perceived quality in the e-commerce context—mainly focused on the website used for a purchase—has been stated as a multi-dimensional construct (Ahn et al. 2007). However, there is not a unanimously accepted version but different contributions including different dimensions and measurement scales of the variable.

In this regard, the research has been mainly targeted at identifying the utilitarian quality which have been confirmed as essential to explain the website quality perceived by users. Different dimensions have been identified, mainly regarding easy-of-use, information quality and customer service (e.g., Kaynama and Black 2000; Madu and Madu 2002; Kim and Lee 2004; Kim et al. 2005; Park et al. 2007). Usability, functionality and security-privacy have been confirmed by Ali (2016) as utilitarian dimensions of hotel website quality.

The hedonic quality has received less attention (Vázquez et al. 2009). First approaches used indirect measurements such as the website design (García and Garrido 2013), a utilitarian attribute that leads back to visual appeal (Park and Gretzel 2007), the level of sociability perceived by users (Barnes and Vidgen 2014) and more recently, the perceived flow (Ali 2016), defined as the development of a pleasant experience, and validated as a mediator variable between utilitarian quality and satisfaction, not as a dimension of website quality. There is a parallelism of these website indicators with the e-store or e-channel attributes identified in other online contexts of study (Ganesh et al. 2010; Betancourt et al. 2017). Nevertheless, the hedonic elements might have increased their importance for digital consumers. The increasing experience and continuous learning of online operations, as well as a more relevant role as a co-producer, are potential determinants to boost pleasant online purchase experiences of the e-customer. Since Park et al. (2007) did not confirm the direct indicators of visual appeal as a determinant of company results, more recent references have provided some advances in this sense. Analysing the online airline tickets purchase context, Llach et al. (2013) find the hedonic dimension of the website quality in the hotel context as a determinant of perceived value. They define hedonic quality as an intrinsic value derived from the enjoyment in searching information and purchasing. The hedonism is measured through five indicators regarding enjoyment visiting the

page, using the information provided, as well as finding the possibility to interact with other users. Although the authors do not propose a second-order variable, their results are enlightening because the hedonic variable and the functional dimension of hotel website quality (based on E-S-QUAL model by (Parasuraman et al. 2005)) were correlated. Ali et al. (2016) extend this model and achieve to confirm the website perceived quality as a second-order latent variable reflected in functional and hedonic dimensions. Ozturk et al. (2016) validate a model that includes utilitarian and hedonic value of the mobile use as direct causes of reuse intentions. The hedonic aspects are measured through three indicators, related to fun and pleasantness in the use of the mobile device.

The second milestone of the value chain is the customer's satisfaction. This is defined as an attitude that deals with an evaluation of the (dis) confirmation of expectations inherent in a product acquisition and/or consumption experience (Oliver 2010). The definition of customer satisfaction varies throughout the marketing literature; however, all definitions agree that the satisfaction implies the necessary presence of a goal that the consumer wants to achieve (Ali et al. 2016).

In the tourism context, satisfaction is considered to be one of the most important results of all marketing activities of market-oriented companies (Kandampully and Suhartanto 2000). Dealing with traditional touristic distribution, contributions such as González et al. (2007); Yüksel et al. (2010); Chen and Xiao (2013) and Deng et al. (2013) relate service quality to expectations, satisfaction and loyalty. In the online context, works such as those by Kim and Lee (2005); Park and Gretzel (2007); and Hao et al. (2015) relate the assessment of the website quality to greater satisfaction with the experience. Llach et al. (2013) discover a positive link between website quality and perceived value, defined according Zeithaml (1988) as the customer judgment or evaluation of the service offered comparing advantages or utility obtained from a product/service and sacrifices or perceived costs. This is a definition close to that of satisfaction, which also requires customer experience (purchase interaction) with the service. Ali (2016) confirms the role of perceived flow as a mediator of the relationship between website quality, satisfaction and hotel rebooking intention.

To the extent that the user perceives a higher quality of an online tourism distribution service, it can be expected to cause an increase in the level of satisfaction with the service.

Moving onto the relationship structure of the chain, the relevance of pursuing the customer's satisfaction is that it leads directly to repeat behaviour and benefits (Ali et al. 2012; Ryu et al. 2012). In addition, it increases the probability of positive recommendations (Oliver 2010). In several research contexts of online distribution channels, it has been shown that greater satisfaction leads to greater intentions of repeated purchases (e.g., Finn et al. 2009; Chiu et al. 2014; Betancourt et al. 2017). Within the context of online tourism, the study of this relationship is a more recent concern, and the results are not conclusive. Even though the relationship between satisfaction and loyalty is postulated as positive, in the model by Bai et al. (2008), the satisfaction of visitors with travel websites is not a mediating variable of the intention to make an online purchase, but there is a direct relationship between the perceived quality of a website and the desire to purchase trips online. The satisfaction measurement deals with the decision to visit a travel website and not with a purchase experience, and the variable for the intention to purchase is related to the virtual travel purchase in general. Kim et al. (2011) find a positive relationship between satisfaction and loyalty. However, once again, the loyalty variable is measured as the intention to purchase at the online tourism channel in general versus a physical store, not as loyalty intention to a brand or company. Amaro and Duarte (2015) call attention to the potential of satisfaction with online purchases of trips to explain the intention to continue using the online option for purchasing trips. They find that attitude and perceived control, which are similar to perceived utility and ease of use according to the authors, are the main determinants of the intention to purchase trips online, which would confirm an attribute relationship between website quality and loyalty. This research does not include the satisfaction variable, and the purchase intention variable is not linked to a specific brand or company. Ali (2016) and Ali et al. (2016) are the only references found in which satisfaction is considered a direct variable of loyalty intentions, particularly in the context of a hotel reservation intention.

Taking into account the revised contributions, it is expected that the higher the customer's satisfaction with the shopping experience provided by an online tourism brand, the greater the customer's intentional loyalty.

Proving that the repurchase of online tourism services is determined by digital tourist' satisfaction is essential; satisfaction and loyalty increments through co-production and co-creation drive higher switching costs (including search costs), which reduce the likelihood to change the provider or the brand of the customer (Jackson 1985). This drives the success of retention strategies and reinforces the long-term survival of the company.

**Hypothesis H1.** *The greater the quality (utilitarian and hedonic) of an online tourism service, the greater the satisfaction with the e-purchasing experience maintained with the online tourist brand.*

**Hypothesis H2.** *The greater the customer's satisfaction with the booked online tourist company, the greater the loyalty intentions towards the brands.*

*2.2. The Role of Perceived Costs*

Mainly due to the increase of consumer participation in online tourism production and distribution processes, the perceived transaction costs from the point of view of the digital consumer may influence notably the relationships between the principal variables pertaining to the value chain, and provide better understanding of the online purchase processes.

In the context of the value chain, transaction costs can play a relevant role as moderating or mediating variables in determining the level of customer satisfaction. According the definition of satisfaction, they would complement the task of the perceived quality. The satisfaction level evaluation requires quantifying a feeling after enjoying a service, that is, a comparison between the benefits received and the costs that are paid (Cronin and Taylor 1992; Berné et al. 1996). Therefore, the perceived transaction costs derived from making an online purchase might be relevant in determining the satisfaction level of an online tourism consumer.

Co-production processes imply the contribution to productivity by means of the consumer participation in the production and distribution of products; particularly in the context of e-commerce, this practice allows for closer relations with consumers (Stockdale 2007). The point is that this new connected, informed and active consumer—announced by Prahalad and Ramaswamy (2004, 2013)—with an intensive role as co-producer, might have to bear higher costs as an operand resource, according to Goods Dominant Logic, and as an operant resource according to Services Dominant Logic (Vargo and Lusch 2004, 2008). The reduction of costs could be an essential trigger for co-creation (Etgar 2006, 2008), and a process aimed at co-creation can lead to lower costs borne by the company, due to the participation of the customer (Ahn et al. 2007).

Shaw et al. (2011) point out that the experiences are the core of the tourism industry and the consumer is the centre of development through co-production processes; thus, providing co-creation opportunities in a consistent way with involvement levels of customer, the experience can be improved and a more consistent competitive advantage can be achieved. Therefore, the digital tourists may prefer higher levels of participation even if they increase their non-monetary costs in exchange for lower prices.

The resources, interests and expectations of both providers and consumers have to be integrated in order to obtain mutual benefits, as a positive effect on market share (Chathoth et al. 2013). In this line, it is essential to analyse the role played by perceived transaction costs in the online booking purchase process.

Transaction costs might be monetary, pertaining to the price of the service, and non-monetary, pertaining to time and effort costs. In our research context, they are perceived costs basically related to the time and the efforts made by the consumer in the purchase process. The Internet allows the

consumers to use the distribution services at any time and place when searching and purchasing a particular product (Betancourt et al. 2016). Hann and Terwiesch (2003) discussed greater cognitive effort and time opportunity when making online purchase decisions, and suggested that the non-monetary costs could turn out to be much more important in co-production activities. Accepting that consumer participation is very dynamic and relevant in online tourism services, another reason arises to consider the role of transaction costs. Nevertheless, despite some contributions, the inclusion of costs in research models has been scarcely considered within the context of online B2C tourism.

Price has a relevant role making purchase decisions (Krishnamurthi and Raj 1988). Price sensitivity has been related to customer satisfaction, so companies with higher satisfaction ratings can set higher prices (e.g., Zeithaml et al. 1996; Cronin et al. 2000). The customers usually consider price as a cue in their expectations of the service performance, which shapes their attitude and behaviour as well (Han and Ryu 2012). Although perceived quality and monetary costs show a positive relationship according to the laws of economics (the greater product quality the greater product price), there is evidence about a negative relationship between costs and customer's satisfaction. In that way, higher perceived costs will drive lower satisfaction. In this line, Jiang and Rosenbloom (2005) observe that favourable price perceptions have direct and positive effects on overall customer satisfaction and on customer intention to return. Han and Ryu (2009) state that customers' perceptions of a reasonable price intervene as a moderator variable to enhance the impact of quality on their satisfaction.

While it is generally accepted that online commerce reduces the transaction costs (Bunduchi 2005), we have not much more knowledge about its role in the value chain. Cho and Agrusa (2006) considered the monetary costs (price) within the specific context of online travel agencies, although the research was limited to studying the variable as a determinant of the website perceived quality. Authors recognize that the price influences the purchaser perceptions both positively and negatively. In their model, price is shown as a determinant of easy-of-use and utility dimensions of the website, and the consumer involvement degree is a mediator variable of the relation. These website quality dimensions determine the attitudes toward online travel agencies, which in turn determine e-satisfaction.

In the same way, Kim et al. (2011) model the transaction costs as antecedents of satisfaction of Korean buyers of online tourism services. In particular, they confirm the monetary transaction costs as direct determinants of satisfaction and also as determinants of indirect effects on trust and loyalty through the satisfaction variable. However, they do not find a direct effect by costs on trust. In comparison with the transaction costs of offline transactions, online monetary costs are positively related to satisfaction. In a hotel context, Ye et al. (2011) find that the perceived price has a determinant effect on quality and a negative impact on the perceived value. Analysing the Spanish online travel agencies, García and Garrido (2013) found a moderate correlation between prices (in monetary units) and some attributes of website (simplicity, clarity and customer service). Authors interpret this result as a greater power of the online channel in order to compare prices. The online environment facilitates comparing prices and consequently, the sensitivity to the price of the customer could increase (Cho and Agrusa 2006).

Regarding the non-monetary costs and their role in the quality-satisfaction-loyalty value chain, there is a lack of knowledge. It is recognized that different type of costs are implicit in the assessment of the perceived quality of a service (Cho and Agrusa 2006), and it is accepted the purchase effort as the perceived difficulty and time costs consumers experience when purchasing a product using a specific channel (Verhoef et al. 2007). Nevertheless, not much more contributions have been found in this respect. The hypothesis will be formulated for overall online perceived costs, and it will defend them as covariates of the level of satisfaction in comparison to other alternatives.

**Hypothesis H3.** *The lower the perceived costs of an online tourism transaction with respect to other (offline and online) alternatives, the greater the satisfaction of the customer.*

The relationships postulated through the formulated hypotheses are included in a theoretical model of the electronic tourist purchasing process (e-TPP model), which involves latent variables of different orders (Figure 1).

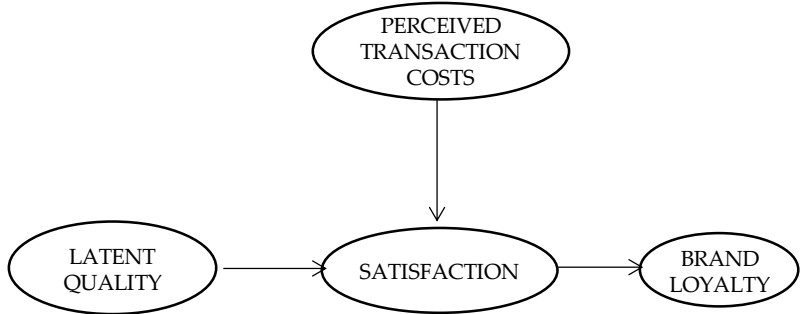

**Figure 1.** E-TPP structural theoretical model.

## 3. Research Methods

A structured questionnaire was developed to obtain the necessary information for testing the working hypotheses and the proposed theoretical model. The target population for this study was limited to those Spanish online tourism services purchasers who have had an online tourism purchase experience over the last twelve months.

*3.1. Measures*

Specific information was requested about the user's experience with the company where they made their last online tourism purchase. The opinion questions were measured through semantic differential scales of 11 points, from 0 for the least favourable option to 10 for the most favourable option regarding the specific proposal (completely adequate, very easy, much less expensive, much less effort, completely satisfactory, yes-always).

The measurement of the perceived quality of a website included the indicators most commonly used in the literature, three of which referred to utilitarian quality (ease of use, information provided by the page, and customer service) and one to hedonic quality (attractiveness of the website). This latter criterion was measured through an overall measure, on a points-scale from very unattractive to very attractive, according Sauro (2015).

Thus, the perceived quality of a website is postulated as a second-order latent variable. Satisfaction is considered a first-order latent variable, measured through one indicator of satisfaction with the last transaction conducted and another indicator of accumulated experience. The perceived transaction costs were measured through three indicators, two of them regarding relative prices (monetary costs) and the third regarding the effort perceived by the user (non-monetary). Loyalty was measured with one item on the recommendation of the service and another on the intention to repeat the purchase (Table 1).

**Table 1.** Criteria, indicators and prior references.

| Criteria, Items | Prior References * |
|---|---|
| Ease-of-use<br>P1. Ease of access to the web page.<br>P2. Possibility offered by the company to combine tourism products in a single order.<br>P3. Perceived clarity (ease of identification) about the company's products and services on its web page.<br>P4. Preciseness (absence of ambiguity) of the definition of the products and services on the company's web page.<br>P5. Purchase payment modes offered through the online service.P6. Task of consumers in the event that they may have combined products in the last transaction or in other, previous transactions with the same company to get the desired combination.<br>P7. Time used to finalise the purchase. | Kaynama and Black (2000);<br>Donthu (2001);<br>Jeong and Lambert (2001);<br>Madu and Madu (2002);<br>Kim and Lee (2004); Kim et al. (2005);<br>Park et al. (2007);<br>Verhoef et al. (2007, buying time);<br>Seiders et al. (2007, accessibility);<br>Jaiswal et al. (2010);<br>Ganesh et al. (2010, ease of payment);<br>Ali (2016, usability) |
| Information attributes<br>P8. The information provided by the company, online, for making the purchase.<br>P9 . . . provided by the company's web page about the characteristics of the contracted tourism service.<br>P10. . . . from the web page about the variety of the online tourism products-services offered by the company. | Kaynama and Black (2000);<br>Jeong and Lambert (2001);<br>Madu and Madu (2002);<br>Kim and Lee (2004);<br>Kim et al. (2005);<br>Park et al. (2007);<br>Verhoef et al. (2007);<br>Ganesh et al. (2010) (merchandise variety),<br>Hung et al. (2014);<br>Ali (2016, functionality) |
| Customer service<br>P11. Confirmation procedure of the booking-purchase, discounts and/or invoices by the company.<br>P12. . . . for cancelling the contracted online tourism service.<br>P13. Customer service and/or complaints and claims system available on the company's web page.<br>P14. Privacy and security policy followed by the contracted online service with respect to the customer's personal data. | Kaynama and Black (2000);<br>Madu and Madu (2002);<br>Kim and Lee (2004);<br>Kim et al. (2005, security);<br>Park et al. (2007);<br>Jaiswal et al. (2010, privacy, security);<br>Ali (2016, security and privacy) |
| Visual attraction<br>P15. Attractiveness of the web page where the tourism service has been contracted. | Kaynama and Black (2000);<br>Kim et al. (2005);<br>Bauer et al. (2006);<br>Urban et al. (2009);<br>Ganesh et al. (2010);<br>García and Garrido (2013) |
| Perceived costs<br>P16. Prices of the service contracted online in relation to purchases in offline channels.<br>P17. . . . with respect to other, similar online services.<br>P18. Effort made in the online purchasing process versus the offline process. | Kim et al. (2011);<br>García and Garrido (2013);<br>Chiu et al. (2014, monetary savings);<br>Verhoef et al. (2007, search and purchase effort) |
| Satisfaction<br>P19. Overall satisfaction with the last purchase of tourism services contracted online (satisfaction with the last transaction).<br>P20. Online purchasing experience over time with the last contracted company (accumulated satisfaction). | Arrondo et al. (2002);<br>Berné et al. (2005);<br>Finn et al. (2009, cummulative);<br>Hung et al. (2014, cummulative);<br>Betancourt et al. (2017, cummulative) |
| Brand/Company loyalty<br>P21. Intention to continue using the same online tourism service.<br>P22. Recommendation of the contracted online service. | Arrondo et al. (2002);<br>Berné et al. (2005);<br>Finn et al. (2009, repurchase);<br>Chiu et al. (2014, repurchase);<br>Betancourt et al. (2017, repurchase) |

* References in parentheses use only the contents contained therein.

*3.2. Data Collection*

A market research company was contracted to distribute the online questionnaire and select a sample (toluna.com) from its e-consumers' panel. The questionnaire is structured in sections according geo-demographic information required and the indicators (Likert scale format) pertaining to the variables included in the model (Table 1). The type of sampling is convenience with quotas, initially requested for an approximation of the specifications observed in literature. The survey conducted by the company attained 408 valid questionnaires, after following a checking process to confirm to the initial requirements. The characteristics of the sample are shown in Table 2.

**Table 2.** Characteristics of the sample.

| | | |
|---|---|---|
| Sex | Male | 50.7% |
| | Female | 49.3% |
| Age | 18–30 | 28.2% |
| | 31–55 | 46.6% |
| | Over 55 | 25.2% |
| Education | Primary education | 1.5% |
| | Secondary education (mandatory) | 6.6% |
| | Higher secondary education | 19.1% |
| | Uncompleted university studies | 10.5% |
| | Higher education (graduated and post-graduated) | 51.9% |
| | Vocational training (post-secondary) | 2.2% |

Regarding the geographic origin of the respondents, the greatest weight by autonomous community corresponds to Madrid (19.6%), followed by Catalonia (18.1%), Andalucía (14.5%) and the Community of Valencia (10%). This matches the population distribution in Spain.

The companies used the most for the online contracting of tourism services by respondents are Booking (14.5%), eDreams (10.8%), El Corte Ingles (10.3%) and Rumbo (7.1%). Booking, eDreams and Rumbo are the online travel agencies named the most regarding the Spanish context in Sarmiento (2016). Odigeo-eDreams and Bravofly-Rumbo led the top 5 of the Hosteltur' Ranking of Online Agencies. Approximately 22% of respondents have contracted the last online tourism service to a supplier (direct channel for hospitality and transport, mainly).

## 4. Results

The data analysis was conducted through Exploratory Factorial Analysis (EFA), Principal Components Analysis (PCA), Confirmatory Factorial Analysis (CFA) and Structural Equation Models (SEM). The literature reviewed supports the content validity of the indicators selected. First, we identify the underlying structure for each of the proposed dimensions (EFA-PCA). Subsequently, the measurement models are validated through CFA. Afterwards, the relationships between the dimensions are tested through SEM (SPSS 22 and EQS 6.0).

*4.1. Ease-of-Use Measurement Model*

After the corresponding PCA with Varimax rotation (PCA-VM) of the first seven indicators (P1 to P7), one component that explains almost the 60% of the variance is obtained. It groups together all the indicators and is called Ease-of-Use (EU) and it refers to the degree of effort that online customers give to the electronic medium (Donthu 2001). It deals with functionality, accessibility of a website, consistency and effective browsing, as well as search capacity and desired products. EU Cronbach's alpha coefficient reaches a value of 0.885 (Nunnally 1978). The estimate of CFA model shows the overall goodness-of-fit statistics and indexes (Table 3). The reliability coefficients of the observed variables take values that exceed 0.5, except for items P1, P2, and P6, whose estimated parameters take values of 0.654, 0.663 and 0.696. These deal with the ease of access and combine products as

well as the task of consumers to reach the desired combination. Due to the importance given to these aspects in literature, we decide to keep them in the model. The most relevant services are the ease of product identification. The values taken by the standardized factor loadings comparing the correlations between factors demonstrates the discriminant validity and the convergent validity of the model. CF1 (Fornell and Larcker's coefficient), and CF2 (Omega's coefficient) take values of 0.536 and 0.889, respectively.

### 4.2. Service Information Measurement Model

Considering P8 to P10, the PCA-VM resulted in a single component that explains 75.5% of the variance. The purchasing experience of customers is increased by the integrity, uniqueness, preciseness and entertainment value of a website, as well as the opportunity for information/content (Kaynama and Black 2000; Aladwani and Palvia 2002; Sigala and Sakellaridis 2004). Thus, service information (SI) can be defined as the degree to which a user believes that the information or content is useful, updated and reliable. Cronbach's Alpha coefficient is 0.862. The corresponding CFA is conducted imposing a restriction on the equality of factor loadings, since this model did not show degrees of freedom (see Table 3). Reliability coefficients of the dimension (CF1and CF2 take values of 0.684 and 0.866, respectively) offer evidence of the reliability and of the convergent validity.

### 4.3. Customer Service Measurement Model

Customer service is determined through the transmission of an appropriate response to e-mail requests or complaints, as well as order confirmations, which represent an important factor in the assessment of a website by customers (see Yang and Jun 2002; Long and McMellon 2004). The dimension may be defined as the desire or willingness for customer service, thereby providing a quick, streamlined service in an online context. PCA-VM confirms the existence of a factor, Customer Service (CS), which explains 64.1% of the variance and includes P1 to P14. Cronbach's alpha takes a value of 0.805. The estimate of the CFA of the CS model presents adequate values of the R-RMSEA statistic and of the goodness-of-fit indexes (Table 3). CF1 and CF2 take values of 0.517 and 0.808. The procedure for cancelling the tourism service receives the lowest value. However, it is kept due to the importance given to this aspect in the literature. The most relevant variable refers to privacy and security policy with respect to the customer's personal data.

### 4.4. Perceived Costs Measurement Model

The PCA-VM offers two components that explain 87.82% of the total variance of transaction costs (PC). The first component groups together the indicators pertaining to the monetary costs borne, and the second one the non-monetary costs. These costs are posed in relation to other situations, either in comparison with offline purchases or other online alternative companies. The correlation coefficient takes a value of 0.777.

The two indicators of perceived monetary costs are grouped into a first-order dimension of perceived online monetary costs (MPC) (Table 4); CF1 and CF2 take values of 0.586 and 0.735, respectively. The other dimension corresponds to non-monetary costs (NMPC).

### 4.5. Satisfaction and Loyalty Measurement Models

Customer's satisfaction (S) is measured considering the satisfaction level with the last transaction (short-term), and the satisfaction with the accumulated experience with the online tourism company (long-term) (e.g., Arrondo et al. 2002; Berné et al. 2005). The underlying dimensional structure for the set of two variables pertaining to the satisfaction, treated through PCA-VM, shows one component that explains 91.94% of the variance. The CFA result shows the adequacy of the identified structure, and the goodness-of-fit statistics and indexes of the model are shown in Table 3.

Similarly, the two variables pertaining to the brand loyalty (L) were subjected to a PCA-VM, and one component that explains 88.56% of the variance was obtained. It concerns attitudinal loyalty

through the intention to make repeat purchases with the online tourism operator and the intention to recommend the operator to other users. CFA estimated presents adequate goodness-of-fit statistics and indexes (Table 3). The CF1 and CF2 values are 0.734 and 0.847, respectively, which indicate reliability and convergent validity.

Both satisfaction and brand loyalty are confirmed as first-order dimensions. Reliability coefficients of the observed variables ($R^2$) exceed 0.7.

**Table 3.** Measurement Models. Goodness-of-Fit.

|     | d.f. | Chi-Square S-B | P      | R-RMSEA | SRMR  | GFI   | AGFI  | R-BBN | R-CFI |
| --- | ---- | -------------- | ------ | ------- | ----- | ----- | ----- | ----- | ----- |
| EU  | 14   | 29.479         | 0.009  | 0.054   | 0.038 | 0.957 | 0.913 | 0.964 | 0.981 |
| SI  | 2    | 0.6012         | 0.740  | 0.0001  | 0.029 | 0.996 | 0.988 | 0.998 | 0.999 |
| CS  | 2    | 9.455          | 0.009  | 0.098   | 0.040 | 0.974 | 0.868 | 0.966 | 0.973 |
| PC  | 2    | 23.872         | 0.0001 | 0.167   | 0.163 | 0.933 | 0.798 | 0.812 | 0.823 |
| S   | 1    | 0.7593         | 0.384  | 0.0001  | 0.028 | 0.997 | 0.992 | 0.996 | 0.999 |
| L   | 1    | 0.0017         | 0.967  | 0.001   | 0.001 | 0.999 | 0.999 | 0.999 | 0.999 |

### 4.6. e-TPP Model

After confirming the different measurement models, the entire structural model was estimated. The results obtained made it advisable to include a previously non-postulated cause-effect relationship between the monetary costs variable and the latent quality variable. We accepted this, given that quality and monetary costs have been previously related in literature (e.g., Cho and Agrusa 2006; Han and Ryu 2009; Ye et al. 2011). After re-estimating the model, the results led to acceptable goodness of fit. The model explains 83.7% of a customer's level of satisfaction. In turn, satisfaction has a positive impact on a customer's attitude of brand loyalty and explains that attitude by 75.2%. CF1 and CF2 coefficients evidence internal validity (see Table 4 and Figure 2).

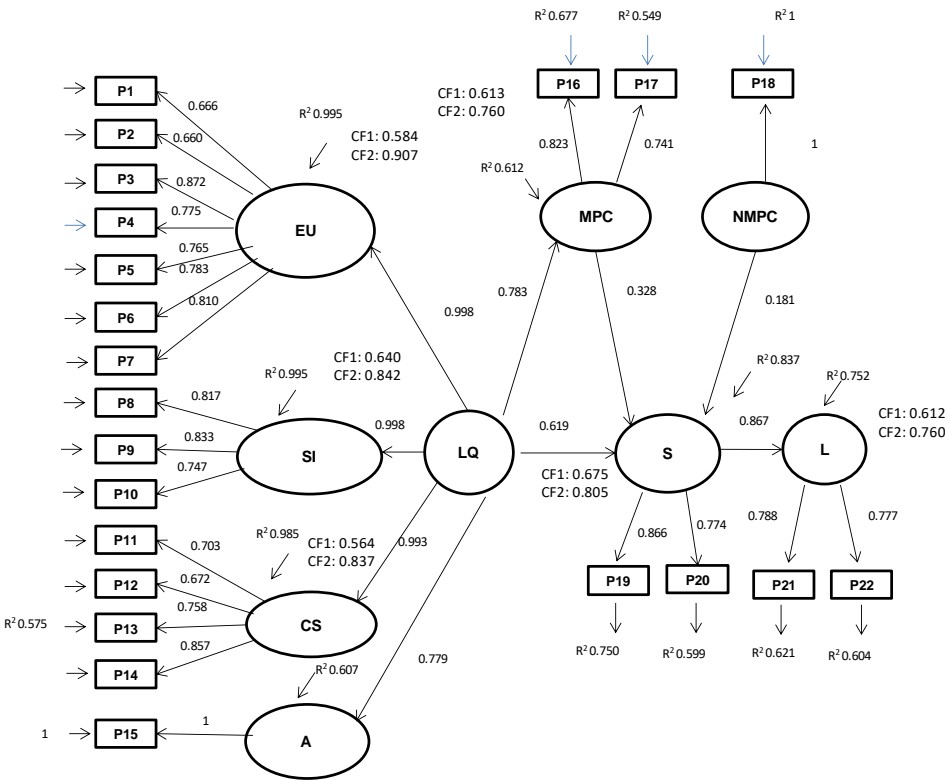

**Figure 2.** Estimation of the e-TPP Model.

**Table 4.** e-TPP Model. Goodness-of-Fit.

|  | d.f. | Chi-Square S-B | P | R-RMSEA | SRMR | GFI | AGFI | R-BBN | R-CFI |
|---|---|---|---|---|---|---|---|---|---|
| e-TPP | 202 | 412.8158 | 0.000 | 0.055 | 0.144 | 0.860 | 0.825 | 0.878 | 0.933 |

Satisfaction turns out to be an effect variable of both the website perceived quality and the perceived costs (monetary and non-monetary). The influence by perceived quality is both direct and indirect through monetary costs (parameters values of 0.619 and 0.257, respectively). The satisfaction variable thus becomes a mediating variable of brand loyalty, unlike the results of Bai et al. (2008).

Monetary costs result in a significant parameter of 0.328 regarding the influence on satisfaction, which indicates that lower perceived prices result in greater satisfaction. The non-monetary costs (0.181) have a lower but significant effect on customer's satisfaction.

Consequently, the hypotheses cannot be rejected. The theoretical model thus reveals a large part of the process of forming attitudinal loyalty to an online tourism brand.

## 5. Conclusions

With respect to the first research question formulated, the results obtained confirm the basic relationships of the quality-satisfaction-loyalty value chain previously found in other study contexts, in the case of B2C tourism online. The e-TTP model explains more than 83% of the satisfaction of digital tourists. While, as a novelty, the value chain is extended, including monetary and non-monetary perceived costs. Monetary perceived costs foster the effect of quality on satisfaction, and non-monetary transaction costs help to determine customer satisfaction. The explanatory power of the model regarding repurchase intentions exceeds 75%. These results give confidence to the comprehensive e-TPP model overcoming some of the limitations of previous models and are in line with Ali (2016) and Ali et al. (2016).

With respect to the second question, this research confirms the important role played by the perceived transaction costs. The consumer participation in the service production, facilitated by online channels, draw a new scenario where the formation of the satisfaction and brand loyalty does not solely involve the website perceived quality. The perceived costs, both monetary and non-monetary, reinforce the customer's satisfaction derived from the relationship maintained with a company, and ultimately, the customer loyalty to the company or brand. It should be noted that the positive relationship between latent quality and monetary costs is contrary to what has been traditionally postulated. The greater the perceived website quality of the tourism service, the greater the perception of a lower price in relation to other channels (online and traditional). Also, the lower perceived price reinforces the customer's satisfaction as well as non-monetary costs. This is relevant because it suggests the existence of an effective win-win dynamic, required according Friesen (2001) to facilitate co-creation by building trust and sharing benefits. The rate presentation might be important as pointed out by Noone and Mattila (2009) and Webb (2016).

Regarding implications, this work provides different measurement models. It is worth highlighting the validated measurement model of the website latent quality in the context of online tourism services purchases. It is reflected in four first-order dimensions: the ease of use, the information provided, the attention perceived by customers through the online environment, and the attractiveness of a website. Even though the attractiveness of a website has less weight in reflecting latent quality than utilitarian dimensions in our research, its valuation by users has a positive influence on the results of the tourism company both in satisfaction and the brand repurchase intentions. It matches the results obtained by Ali et al. (2016) and Ozturk et al. (2016), and is in line with Betancourt et al. (2017) for online private sales clubs, where the attractiveness of the design had an influence on loyalty for the most-satisfied group of customers.

In this sense, future research about perceived website quality should include the hedonic dimension. For practitioners, the hedonic content of the website could be an element of differentiation

and an instrument for achieving useful competitive advantages in the acquisition and retention of customers.

Moreover, this research shows that multi-dimensional constructs increases the overall construct understanding (Law et al. 1998), and thereby provides details about its various facets (Petter et al. 2007). Thus, tourism managers can use, with guarantee, the validated instrument to analyse the situation of each website quality dimension in order to identify those aspects that may need improvement. The structure of latent quality can be considered to support decision-making taking into account different types of customers and characteristics of the channel or market where the company is operating.

In addition, the opportunity to interact with a customer must be taken as an advantage ensuring that perceived costs are included in the development of strategies for customer acquisition and retention. The way in which perceived monetary costs are handled can differentiate a company, considering that an improvement in website quality from the user's point of view is not related to the acceptance of higher prices for a tourism service, rather it has the opposite effect. Therefore, a company must find justification for prices through more efficient processes, both internal (such as centralising the control of distribution) and external (such as through the negotiation of commissions or selecting the channels in which commissions are lower). Care must be taken with non-monetary costs by providing efficiency for the user, whose participation in the purchasing process depends on lower effort in the virtual interaction.

Despite the highlighted contributions of this research, some limitations have to be taken into account, thereby serving as proposals for future research. First, our model is tested on a sample of convenience, so that the level of representativeness of the sample can be affected. This situation is addressed using quotas, although the success is dependent on the accuracy of the selection made by the contracted market research company. In addition, despite the high explanatory power of the model, it could be reinforced by adding control variables, such as the personal characteristics of customers and specific conditions of the market under analysis or loyalty of behaviour. On the other hand, given that the study explores a situation that groups together all possible travel contexts, it could be of interest for research to specify online purchasing situations for different purposes. Also, to explore the dimensionality of the hedonic character of the quality of the website in function of the different situations of purchase and the different supports or screens used by customers is an interesting line of future research. In addition, the analysis of the longitudinal databases available to companies should allow them to make comparisons over time as a result of eventual changes in the variables.

In the academic scope, inter-cultural comparisons could offer reinforcement of the theoretical model and offer results that allow justifying the implementation of differentiated marketing strategies. Studying the behaviour of the e-TPP model in different types of online tourism channels is another task for future research. A comparative study of actions in the direct channel and the indirect channels could better illustrate the situation of competition in the sector from the user's point of view. Furthermore, in addition to the online tourism channels observed in this work and in the tourism distribution system, there are channels whose development is currently growing such as the channel consumer-to-consumer (C2C), and social networks, although these are still marginal as a sales channel according Stangl et al. (2016).

In any event, this research contributes a holistic model that goes beyond others by including variables that have been confirmed relevant as mediators of brand loyalty. The confirmed model manages to integrate the phases of the online tourism purchasing process and shows how the development of the business-client interaction is achieving the objectives of both. The work provides a meaningful explanation of the purchasing process and the results of online tourism companies, thus helping to implement strategies to acquire and, above all, retain brand customers.

**Author Contributions:** Conceptualization, C.B.-M. and M.P.-I.; Methodology, C.B.-M. and M.M.-N.; Validation, M.M.-N.; Investigation, M.G.-C.; Writing-Original Draft Preparation, M.G.-C. and C.B.-M.; Writing-Review & Editing, C.B.-M.; Supervision, C.B.-M. and M.P.-I.

**Funding:** We acknowledge the support for this research from the Crevalor Research Group and the UZ2018-SOC-04 Research Project.

**Conflicts of Interest:** The authors declare no conflict of interest.

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
