# Peer review of "Reviewing the Online Tourism Value Chain"

_admsci, doi:10.3390/admsci8030048_

Round 1

Reviewer 1 Report

In this paper is to build a model which is able to measure online purchasing process. And this process is affected by before and after purchasing customer behavior that is related to how booking process in e-tourism is success or not. Therefore, the online tourism value chain (quality-satisfaction-loyalty) is development for exploring digital tourism customer how to work smoothly in the designed website. This proposed model is namely e-TPP model which is connected by electronic tourist purchasing process within different measured variables.

The proposed model is examined by collected data from Spanish online tourism services purchasers who have had used experiences in the booking websites, where has online purchasing happened or transaction during last twelve months. The results are supported by four different analyzed approaches such as exploratory factorial analysis (EFA), principal components analysis (PCA), confirmatory factorial analysis (CFA) and structural equations models (SEM).

This is a well-written paper since it provides enough information and background knowledge. The paper provides an important observation for online tourism value chain as well as shows a practical framework.

Only one suggestion for this paper is the Table 1 is not easy to understand since the provided references are not one by one to fit considered criteria or items. For example, the item P1 - Easy of access to the web page is fit one reference or all references. 

Author Response

Thank you for your comments and encouragement. 

We have added in Table 1 an explanation of the brackets in order to clarify the role of prior references used in our survey.

We have reviewed the entire document in order to solve grammatical errors.

Sincerely yours.

The authors.

Reviewer 2 Report

Dear authors,

Your research is very interesting. However, I recommend you to explain:

  - the representativeness of the sample surveyed

- to quote articles from the magazine to increase the value of the magazine in which the article will be published.

Much success in future research!

Author Response

Thanks for your comments and recommendations.

In this sense, we have added a sentence in Conclusions to clarify the level of representativeness of the sample surveyed.

The Administrative Sciences journal has not published at the moment any investigation in the line covered by the current work. Therefore, we are not able to quote a direct contribution. However, we understand your suggestion, and we have quoted one reference from the Administrative Sciences journal, regarding the collateral matters and trying to collaborate in increasing its value.

Sincerely yours,

The authors.

Reviewer 3 Report

Initially I was excited with regards to the title as Transaction Cost Economics (TCE) are are particularly relevant and interesting ín the e-Tourism context.  Over the years, they have been used to explore the overall discussion of internet-enabled dissintermediation in the tourism sector.  The author(s), use the term- transaction costs in a rather different manner, which better correspond to perceived usefulness (see Technology Adoption Model).  So in this sense the term Trasaction costs seems misplaced and misleading to me.  The model constructed follows the logic of an adapted Technology Adoption Model, which has been extensively applied in e Tourism.  The author's model follows exactly the same same logic, just with re-worded terms.  So in sense, the reader is left with the feeling of a 're-invented wheel'.  In terms of methodology, the data collection process was outsourced to a marketing company - which is in principle fine - but there is very little information and transparency on sample selection, quest. design and scales, etc.  Transparency is a key component of a sound methodology!  Another issue here is that the term e-tourism is rather simplistically scoped and superficially defined, basically being limited to online travel agencies (selling what? Hotels, flights, packages, excursions?).  The online distribution chain in tourism entails a number of different holiday components (with different levels of perceived risk and information complexity) and also involves B2B aspects.  The language needs some editing too (one does not 'hire' a holiday...  They 'book it').  All in all, this may be interesting for someone with no background in tourism, but it needs significant work in terms of conceptual clarity and theoretical scope.  The theoretical foundations need some major reqork.  Under those circumstances, I would normally decide to reject the paper, but given the complexity of the topic and the generalised focus of the journal, I recommend major revision.  All the best success and luck to the author(s)

Author Response

1. “… Initially I was excited with regards to the title as Transaction Cost Economics (TCE) are are particularly relevant and interesting ín the e-Tourism context.  Over the years, they have been used to explore the overall discussion of internet-enabled dissintermediation in the tourism sector.  The author(s), use the term- transaction costs in a rather different manner, which better correspond to perceived usefulness (see Technology Adoption Model).  So in this sense the term Trasaction costs seems misplaced and misleading to me.  The model constructed follows the logic of an adapted Technology Adoption Model, which has been extensively applied in e Tourism.  The author's model follows exactly the same same logic, just with re-worded terms.  So in sense, the reader is left with the feeling of a 're-invented wheel'…”

We agree. The transaction costs we use are perceived transaction costs. They are transaction costs from the point of view of the customer-consumer. In marketing, they are very important in order to explain consumer behaviour and distribution channels research (Betancourt, 2004; Berné, 2006). The paper tries to explain the gap observed in the literature in this sense and how the current research tries to fill it.

2. “…In terms of methodology, the data collection process was outsourced to a marketing company - which is in principle fine - but there is very little information and transparency on sample selection, quest. design and scales, etc.  Transparency is a key component of a sound methodology! …”

Information on these topics is presented in the Research Methods section, pages 6, 7, 8 and 9. We have reviewed this section and added more information in order to cover the omissions detected.

About the reliability of the contracted company we can say that it is a company with acknowledged reputation of excellence at international level, and with which the authors accumulate a satisfactory experience of collaboration. The company also has a positive word of mouth from other research groups that are also its clients. In any case, it is an outsourced work that has a monetary economic cost that must correspond to a good execution of the requested service.

In this regard, we trust that the selection of the sample was made following our instructions: selection of people from the company's data panels who complied with the requirements of having made a purchase in the last 12 months and who were able to identify the company last contracted and what services provided, as well as identifying other companies in which to make purchases: online tourism bookings. In addition, geo-demographic quotas were requested for the sample to be as representative as possible of the social situation in Spain. All this is explained in the text.

The measurement scales are explained in the sub-heading of Measures, page 7.

The design of the survey is explained in the sub-heading of data collection, page 8. Some words have been added to clarify and extend the explanation as suggested.

3. “… Another issue here is that the term e-tourism is rather simplistically scoped and superficially defined, basically being limited to online travel agencies (selling what? Hotels, flights, packages, excursions?).  The online distribution chain in tourism entails a number of different holiday components (with different levels of perceived risk and information complexity) and also involves B2B aspects…”

The quality-satisfaction-loyalty chain deals with B2C contexts, between companies and the end customer. Taking into account the context, when we use the term of electronic tourism, we refer to online relationships between digital tourists and companies. Anyway, we have changed the term e-tourism to B2C tourism online. In this way, we hope that the use of the term does not lead to erroneous interpretations.

We fully agree that e-tourism includes analysis, design, implementation and application of IT in the travel and tourism industry. Therefore, it includes travel agencies, suppliers, other intermediaries (meta-search engines, search engines, wholesalers and retailers), and tourists (as end customers). However, regarding the distribution chain, options such as B2B, C2C, C2B2C are not research objectives in the current work, which focuses on electronic tourism from the point of view of digital tourists, analysing consumer markets, not industrial markets That is, our objective is to analyse the relationship between the business and the consumer (B2C, being B suppliers and intermediaries, and C the digital tourist), the last transaction of the chain.

Hotels, flights, packages, excursions, contracted-booked-purchased online, could be an option for the respondents. We have added in the text the percentage (22%) of responses pertaining to this case.

4. “… The language needs some editing too (one does not 'hire' a holiday...  They 'book it')…”

The document has been reviewed by an English’ native professional in order to solve grammatical or other language errors.

5. “… All in all, this may be interesting for someone with no background in tourism, but it needs significant work in terms of conceptual clarity and theoretical scope. The theoretical foundations need some major reqork.  Under those circumstances, I would normally decide to reject the paper, but given the complexity of the topic and the generalised focus of the journal, I recommend major revision…”

We regret having given this impression. We have reviewed the complete document and tried to clarify the aspects that can give conceptual confusion or basic theoretical omissions. We appreciate your understanding and your suggestions, which have undoubtedly led to a substantial improvement of the manuscript.

6. “…All the best success and luck to the author(s)…”

Thank you.

Sincerely yours,

The authors.

Round 2

Reviewer 3 Report

The authors successfully attempted to address the raised issues of my review.  Yet the conceptual issues are still present.

#s3gt_translate_tooltip_mini { display: none !important; }